# Assessment and Learning in Knowledge Spaces (ALEKS) Adaptive System Impact on Students' Perception and Self-Regulated Learning Skills

**Hoda Harati [1,*], Laura Sujo-Montes [1], Chih-Hsiung Tu [1], Shadow J. W. Armfield [1] and Cherng-Jyh Yen [2]**

[1] Department of Educational Specialties, Northern Arizona University, Flagstaff, AZ 86011, USA; Laura.sujo-montes@nau.edu (L.S.-M.); chih-hsiung.tu@nau.edu (C.-H.T.); Shadow.Armfield@nau.edu (S.J.W.A.)

[2] Department of Educational Foundations and Leadership, Old Dominion University, Norfolk, VA 23529, USA; Cyen@Odu.edu

\* Correspondence: Hoda.Harati@nau.edu

**Abstract:** Adaptive learning is an educational method that uses computer algorithms and artificial intelligence (AI) to customize learning materials and activities based on each user's model. Adaptive learning has been used for more than 20 years. However, it is still unique, and no other system could bring more or even similar capabilities than the ones adaptive technology offers, including the application of AI, psychology, psychometrics, machine learning, and providing a personalized learning environment. However, there are not many studies on its practicality, usefulness, improving students' learning skills, students' perception, etc., due to the limited number of institutes investing in this new technology. This paper presents the results of administering the newly developed Adaptive Self-regulated Learning Questionnaire (ASRQ) in an adaptive learning course equipped with the ALEKS (Assessment and Learning in Knowledge Spaces) system to study the amount of Self-regulated Learning Skills (SRL) score change, if any, of the students. The ASRQ was administered at the beginning and end of the semester as a pretest and posttest. Then, the quantitative Sample Paired *t* Test was run to measure the students' SRL score change between the beginning and end of the semester. The results showed a significant decline in students' SRL skills score while working with ALEKS. This paper also discusses the reasons for the considerable drop in SRL skills based on students' perception and feedback collected through administering an open-ended survey at the end of the semester. The survey's qualitative analysis showed various possible factors contributing to the decline of the SRL skills score, including lack of motivation, system complexity, hard penalty, lack of social presence, and lack of system practicality.

**Keywords:** adaptive learning; self-regulated learning; dependent *t* Test; adaptive Self-regulated Learning questionnaire and survey

## 1. Introduction

The Internet has become a powerful tool in the field of education via creating versatile online learning for everyone at any time and place. In recent years, the focus on online learning has led to a more flexible, web-based, and interactive learning environment. The importance of online learning is felt more than ever before due to the recent Coronavirus Disease 2019 (COVID-19) pandemic, which led to the closure of in-person education and caused serious problems for universities [1]. Some of these issues were overcome by applying the information and communication technologies with online applications, multimedia, and courseware. However, all these learning platforms have not been specifically designed to address the individual needs of each learner [2]. Adaptive learning, as a new form of online learning, can focus on the needs of each learner and help them with their learning process, adapting the learning materials based on their knowledge level, and

personalizing their learning path when there is no direct access to in-person instructors. Adaptive learning, known as the next generation of online learning [3], is an alternative to the one-size-fits-all through creating an individualized curriculum. It is also an approach to a personalized learning experience [4]. Adaptive learning is a pedagogical initiative based on analysis, by the academic environment, of data generated during the students' learning process [5]. However, implementation of this pedagogical initiative has its own challenges, such as a new courseware environment with different procedures, instructions, or assessments; absence of learners' social presence; time consuming; and being used in traditional 16-week semesters regardless of students' pace of learning, to name a few [6]. Therefore, this valuable but complicated learning environment requires autonomous learners to be equipped with techniques, skills, and strategies to incorporate different resources into their learning behavior. An important set of skills that learners need to possess is Self-regulated Learning (SRL) [7,8]. Numerous studies [9,10] showed the importance of SRL skills in the success of students in online settings because they can help learners construct knowledge, complete tasks, and, therefore, improve performance. In online adaptive learning, similarly, high SRL skills can assist learners in managing electronic resources, using them as planned, integrating them into their performance and, finally, applying them in their workplace [8]. However, no research has investigated the different roles of SRL skills in adaptive learning environments. Additionally, the previous studies regarding adaptive learning have usually been administered in non-academic settings or non-degree-based majors. Some of these studies were designed for a limited group of students in which an adaptive system was developed specifically to tailor the needs of that particular group. This study is significant because it was administered at the university level with a very known system, i.e., ALEKS. Due to the importance of SRL skills, this research planned to investigate the role of the ALEKS adaptive system in changing SRL skills in students and how they reflected on their experience of using this system.

This article is organized as follows: Section 2 reviews the purpose of this research; Section 3 describes the theoretical framework; Section 4 shows the materials and methods; Section 5 shows the results; Section 6 discusses the results obtained; Section 7 presents the implications; Section 8 proposes the suggestions for further research; and Section 9 conclusions.

## 2. Purpose

The purpose of this study was to identify how much an adaptive learning system impacted students' total SRL skills score in eight variables, including task strategy, perception, goal setting, persistence, self-evaluation, time management, environmental-structuring, and help seeking. Additionally, this study intended to understand the students' feelings, perceptions, and reflections when using the system as their primary courseware. Therefore, it empirically investigated one of the most used adaptive learning systems (i.e., the Assessment and Learning in Knowledge Spaces (ALEKS)) in higher education institutes through the following research questions:

1.   Does the students' total self-regulated learning skills' score in eight SRL variables (task strategy, perception, goal setting, persistence, self-evaluation, time management, environmental-structuring, and help seeking) change during a semester when using the ALEKS adaptive learning system?
2.   What are the students' perceptions of using ALEKS as their adaptive learning system?

To answer the first research question, the Likert scale Adaptive self-regulated Learning Questionnaire (ASRQ) [11] was used to measure the students' SRL score changes during an academic semester (16-week semester). This questionnaire focused on eight SRL variables including task strategy, perception, goal setting, persistence, self-evaluation, time management, environmental-structuring, and help seeking. The ASRQ was implemented twice, at the beginning and end of the semester.

To answer the second research question, a self-reported, open-ended survey was developed and implemented at the end of the semester to determine the students' percep-

tions of ALEKS adaptive learning system. The context of the course was "Introduction to Chemistry" that was taught using ALEKS as the courseware to freshman and sophomore Chemistry students. ALEKS was a mandatory course shell delivered and supported by the corresponding university.

The following section aims to give a background to the main terms of the study, including self-regulated learning skills, adaptive learning, and self-regulated learning in the adaptive learning environment. Then, it continues with the method and the research design used within the methodology. The research design section is divided into data collection, population, instruments including ASRQ, survey, and ALEKS system, and followed by data analysis containing methods used to interpret and validate the data. The article then concludes with the results, discussion, implication, future research, and conclusion.

### 3. Theoretical Framework

This section focuses on the definition and background of SRL skills, adaptive learning, and also implication of SRL skills in adaptive learning environments.

#### 3.1. Self-Regulated Learning Skills (SRL)

SRL is defined as the learner's ability to engage independently and autonomously in a self-motivating and behavioral process fortifying goal achievement [12]. It is an "active constructive process, whereby learners set goals for their learning and then attempt to monitor, regulate, and control their cognition, motivation, and behavior guided and constrained by their goals and the contextual features in the environment." [13] (p. 453). SRL is an umbrella term that covers variables such as cognition, meta-cognition, behavior, motivation, emotion [14], resource management, goal setting, success expectations, and deep cognitive involvement factors such as influencing on learning and helping learners to face and resist under challenging tasks [8,15]. SRL is a learner's complex metacognitive effort to monitor, manipulate, improve their learning [8], and make appropriate decisions when needed [16]. Self-regulated learners are responsive to self-oriented feedback about learning effectiveness and are involved in self-motivational processes [17]. In short, SRL is a developmental cycle that appears to be a function of the interaction among personal, behavioral, and environmental factors [6], [17–19].

The ASRQ instrument used for this study contained eight SRL variables. These eight variables included task strategy, perception, goal setting, persistence, self-evaluation, time management, environmental-structuring, and help seeking. Table 1 presents the definitions of these SRL skills (adopted from [20]) which were modified based on the general features of the adaptive learning system (ALEKS) in this study.

**Table 1.** SRL Skills Definition.

| SRL Skills | Definition |
| --- | --- |
| Task strategy | The learner strategies to tackle adaptive learning systems' complexities to complete tasks |
| Perception | The reflection of learners on their emotions and experiences throughout the learning process |
| Goal setting | The self-initiated plan making based on adaptive learning system's instructions |
| Persistence | The learners' efforts to accomplish adaptive learning materials |
| Self-evaluation | The tracking of progress, success, failure, topics completed, or topics remaining based on the system evaluation graphs |
| Time management | The learners' time set aside for tasks based on the system timetable and instructions. |
| Environmental-structuring | An adaptive learning system's dashboard arrangement to make it more favorable to pursue learning objectives |
| Help seeking | The self-initiated knowledge resource-seeking for better understanding of adaptive learning systems' objectives |

### 3.2. Adaptive Learning

The concept of adaptive learning is to adjust the learning environment's characteristics to meet each learner's individual needs using student-generated data to tailor the path and pace of learning [16]. With the rise of Artificial Intelligence in the 1970s, adaptive technology continued to grow slowly. Adaptive technology was first used in NASA and in the U. S. military for simulation of training models. The Army Adaptive Learning Model was among the first approaches that assisted instructors, training developers, and quality assurance developers to switch from traditional methods to more modern practices of new concepts [21]. Later on, the Department of Education endorsed adaptive technology to simultaneously teach and assess students in different subject matters [22]. Nevertheless, it was not until 2011 that adaptive technology was used more profoundly in higher education institutions; its use was accelerated in 2015/2016 more than ever before [3].

It is essential to distinguish the three main concepts around adaptive learning, i.e., adaptive learning, adaptive learning technology, and personalized learning. Adaptive learning is considered a method to deliver personalized learning to offer efficient, effective, engaging, and customized learning pathways for each learner. Additionally, adaptive technology is the system or courseware paving its way to become the primary educational technology tool to serve the educational practices of personalized learning [3]. Adaptive learning technology tailors the courses and curricula in response to the individual needs of every learner. This technology uses a data-driven approach to adjust the path and pace of learning, enabling the delivery of personalized learning [23]. Personalized learning is a way of teaching and learning in which the course experience is fine-tuned based on the learner's individual needs. In short, adaptive learning is a form of personalized learning in which adaptive learning technology plays a major role [3].

Adaptive learning systems are helpful for learners with different interests, knowledge, or goals. Knowing the learners' level of knowledge, the system can support them in their navigation by "limiting browsing space, suggesting more relevant links to follow, or providing adaptive comments to visible links." [24] (p.1). The system collects all users' information, such as preferences, knowledge level, need, goal, right or wrong answers, length of time in making decisions, and individual strategies throughout the learner's interaction with the system; it then uses the information to adapt concepts, materials, learning pathways, and assessments in order to provide a personalized learning experience [22,24] for each learner. As the traditional one-size-fits-all curricula do not support the individual needs of learners in online settings [25], the adaptive learning system employs Artificial Intelligence to provide a possible approach to such experience [26]. The adaptive learning system supports the student-driven acquisition of the learning material [24]; works as constant tutoring that checks the understanding of the learners in real time [27]; identifies the psychological sources of errors in students' performance [28]; provides intelligent feedback to students; and helps them to reflect and think about their mistakes.

### 3.3. Self-Regulated Learning in Adaptive Learning Environments

Different studies show that self-regulated learners are more successful in their academic achievement. For instance, learners with higher SRL skills in online and blended learning show more positive learning achievement than learners with lower SRL [29]. Furthermore, the use of SRL skills would lead to positive online learning and online learners with higher SRL skills demonstrate better learning performance [29]. SRL skills are fundamental in online settings where learners should demonstrate independence and self-autonomy to learn the materials and interact with others [30]. Adaptive learning, as a form of online learning, is not an exception. In adaptive learning, there is a vital need to develop autonomous, independent, and self-regulated learners to survive in the individualized environment of adaptive systems. In online-only adaptive learning environments, students interact only with the system and do not have any communication with others. Therefore, there is a need to teach SRL skills to students well in advance of engaging with systems and to allow them to enhance these skills to work with this technology independently. This

technology draws on different applications, files, databases, adaptive content in multiple shapes, features, modes, and various forms such as webpages, texts, and graphics [24] to create a complicated learning environment for new users. To deal with this complex system, learners need unique skills to learn content and materials. Learners without SRL skills might face problems managing adaptive resources, incorporating them into their prior knowledge, optimizing their performance, managing their time, monitoring, and reflecting on their performance. To better respond to this wide range of requirements, the adaptive learning systems need to have the capability to develop autonomous learners with high SRL skills who are able to manage their learning process autonomously and use cognitive, metacognitive, and motivational behavior systematically to gain the predetermined academic course goals. Therefore, this study intended to explore if the adaptive learning system currently in use in the studied institution (ALEKS, in this case) helps students to develop their SRL skills.

## 4. Materials and Methods

### 4.1. Participants

The participants of this study were undergraduates in the Introduction to Chemistry adaptive learning course who voluntarily agreed to take part in this institutional review board-approved research. The participants were pursuing a Bachelor's Degree in Chemistry Engineering at a Southwestern University in the USA. The course was conducted over 16 weeks and included two in-person sessions and one online adaptive session per week. The online part of the course was instructed through ALEKS and was graded separately from the in-person one. Three hundred students ($n = 300$) enrolled in this course were invited to participate in this study and fill out the ASRQ and survey. Out of all the invited students, 120 completed the research requirements and were included in this study ($n = 120$). Thus, the study's response rate was 40%. The invitation to participate and the consent form were sent to the students by the course instructor. The consenting participants took part in the study and filled out the questionnaire twice, at the beginning and the end of the semester. They also completed the open-ended survey at the end of the semester. Most of the participants ($n = 120$) were female ($n = 98$, 81.7%), white ($n = 81$, 67.5%), and aged between 18 to 24 years old ($n = 118$, 98.3%). Furthermore, more than half of them were freshmen ($n = 66$, 55%). The detailed demographic information of the participants is shown in Table 2.

### 4.2. Instruments

#### 4.2.1. ALEKS Adaptive Learning System

As an online assessment and tutoring service, ALEKS is an adaptive learning system designed, developed, and commercialized by McGraw Hill. ALEKS was designed mainly for Math and Science courses, but it is now available for K-12 and other Higher Education courses. After logging into ALEKS for the first time, students have access to tutorials and a guided tour of key features of the system, how they can navigate in the system, and how they can succeed. After completing the tutorial, students are directed to take the initial knowledge check to determine what they know, what they do not know, and their knowledge level for the given subject. ALEKS periodically prompts students to complete progress knowledge checks to monitor knowledge, learning retention, and confirm mastery of topics [31]. ALEKS determines the students' knowledge level by asking a few questions to engineer their learning path in the shortest possible way. At the university that this study was performed, two adaptive learning systems were available. However, due to the larger number of students using ALEKS, this system was selected for the purpose of this study and assessing the usefulness of this system in improving SRL skills.

**Table 2.** Demographic Information of Participants (*n* = 120).

| Variable | Frequency | % |
|---|---|---|
| Gender | | |
| Female | 98 | 81.7 |
| Male | 22 | 18.3 |
| Ethnicity | | |
| White | 81 | 67.5 |
| Hispanic or Latino | 4 | 3.3 |
| Two or more races | 14 | 11.7 |
| Middle Eastern or Asian | 12 | 10 |
| Black or African American | 2 | 1.7 |
| American Indian or Alaska | 5 | 4.2 |
| Native Hawaiian or Pacific Island | 2 | 1.7 |
| Age | | |
| 18–25 | 118 | 98.3 |
| 26–35 | 2 | 1.7 |
| 26–36+ | 0 | 0 |
| Grade | | |
| Freshman | 66 | 55 |
| Sophomore | 38 | 31.7 |
| Junior | 13 | 10.8 |
| Senior | 3 | 2.05 |

Note: Descriptive data analysis of the participants attended the Pre-Posttest.

### 4.2.2. Adaptive Self-regulated Learning Questionnaire (ASRQ)

The Adaptive Self-Regulated Learning Questionnaire contains 25 Likert scale items ranging from "Strongly Disagree" (=1) to "Strongly Agree" (=5), focusing on eight SRL skills including goal setting, environmental-structuring, time management, task strategy, self-evaluation, help seeking, persistence, and perception as previously explained in Table 1. The entire ASRQ, with questions related to each of the eight factors, is in Appendix A. The reliability and validity of the ASRQ were established via the review of a content expert panel, Cronbach's alpha coefficients, and confirmatory factor analysis [11]. Overall, the results supported a reliable (*a* = 0.94) and valid (CFI = 0.90) instrument to measure SRL skills in an adaptive learning environment [11]. For this research, the questionnaire was administered at the beginning as the pretest and at the end of the semester as the posttest.

### 4.2.3. Open-Ended Survey

The open-ended survey containing eleven open-ended questions (Appendix B) was distributed among the participants at the end of the semester to gather information about their perception toward ALEKS and their SRL skills. The content validity of the survey was evaluated in the pilot test of the instrument by three experts in the field who checked the relevance, clarity, simplicity, and ambiguity of the survey. Additionally, the reliability of the responses was determined through the triangulation of the data through coding the survey responses by two coders, which is explained more depth in the data analysis section.

### *4.3. Data Analysis*

Data analysis in this study contained both quantitative and qualitative analyses. The quantitative data analysis was run via IBM SPSS Statistics 24. Furthermore, the qualitative part of this research study was conducted through a thematic analysis of the texts obtained from the open-ended survey. Each of these analyses is discussed as follows.

### 4.3.1. Qualitative Analysis of the ASRQ

Three quantitative tests were run on the data gathered from the ASRQ to answer the first research question. After collecting data from the ASRQ and matching the students'

pretest with their posttest, data were fed into the SPSS 24. First, the Shapiro–Wilk Tests of Normality was run. Then, for each pretest and posttest, the total mean score of SRL skills in eight variants (including task strategy, perception, goal setting, persistence, self-evaluation, time management, environmental-structuring, and help seeking) was calculated. Next, to determine whether there was a significant difference in the SRL skills score from the beginning toward the end of the semester, a dependent $t$ test for paired samples was implemented using 0.05 as the significance level.

### 4.3.2. Qualitative Analysis of the Survey

A qualitative analysis was used in this study to identify the themes of each survey question and the percentage of each theme. For this purpose, two coders were employed, and a coding orientation was provided to help them follow the same procedure. The coders read, categorized, coded, and organized the texts gathered from each survey question based on the given instructions. The two coders' themes were compared and the discrepancies between the themes were discussed, justified, and fixed by the researchers. Then, the percentage of each theme in each question was calculated. The inter-rater reliability ($\kappa = 0.87$) showed high consistency of the measurement between the two coders. The detailed discussion of the results of qualitative analysis and the themes of each question are discussed in part 5.2.

## 5. Results

### 5.1. Tests of Normality

The Shapiro–Wilk Tests of Normality ($0.147 > p$) was run, and it indicated that the result was not statistically significant. Therefore, the data were normally distributed (Table 3).

**Table 3.** Test of Normality.

| | Kolmogorov–Smirnova | | | Shapiro–Wilk | | |
|---|---|---|---|---|---|---|
| | *Statistic* | *df* | *Sig.* | *Statistic* | *df* | *Sig.* |
| Differences | 0.054 | 120 | 2000 * | 0.983 | 120 | 0.147 |

Note. * This is a lower bound of the true significance.

### 5.2. Descriptive Statistics of the Variables

The descriptive statistics of the SRL variables in pretest and posttest are listed in Tables 4 and 5. Overall, participants had a higher level of self-regulation in the pretest with a total Mean Score of (M = 88.07). Participants seemed to have stronger 'perception' and 'task strategy' skills but weaker 'environmental-structuring' and 'help seeking' skills in both pretest and posttest.

### 5.3. Dependent Test for Paired Samples

The total SRL skills' scores for the pretest and posttest were used to determine whether students' overall score changed from the beginning to the end of the semester while working with ALEKS. The dependent $t$ test analysis showed that the difference between pretest and posttest SRL scores was significant. However, the pretest ($M = 88.07$, $SD = 8.07$) reported significantly higher SRL skills than posttest ($M = 83.22$, $SD = 7.22$, $t$ [119] = 4.178, $p < 0.05$). This result emphasized that the overall SRL skills' score of the students who participated in this study and used ALEKS as their adaptive system to learn Chemistry dropped significantly at the end of the semester compared to their score at the beginning of the course; therefore, ALEKS was not effective in increasing students' SRL skills' scores (Tables 6 and 7).

**Table 4.** Descriptive Analysis of Pretest (*n* = 120).

| Variables | Number of Survey Items | M | Range | SD | Min | Max |
|---|---|---|---|---|---|---|
| Goal setting | 3 | 11.01 | 12.0 | 2.47 | 3.0 | 15.0 |
| Environmental-structuring | 3 | 9.16 | 12.0 | 2.67 | 3.0 | 15.0 |
| Task strategy | 4 | 13.63 | 16.0 | 3.45 | 4.0 | 20.0 |
| Time management | 3 | 9.93 | 12.0 | 2.75 | 3.0 | 15.0 |
| Help seeking | 3 | 9.5 | 12.0 | 2.41 | 3.0 | 15.0 |
| Persistence | 3 | 11.63 | 12.0 | 2.71 | 3.0 | 15.0 |
| Self-evaluation | 3 | 10.26 | 12.0 | 2.65 | 3.0 | 15.0 |
| Perception | 4 | 12.94 | 16.0 | 4.36 | 4.0 | 20.0 |

Note. Survey items were constructed with a 5-point Likert scale ranged from 1 strongly disagree to 5 strongly agree.

**Table 5.** Descriptive Analysis of Posttest (*n* = 120).

| Variables | # of Survey Items | M | Range | SD | Min | Max |
|---|---|---|---|---|---|---|
| Goal setting | 3 | 11.03 | 12.0 | 0.26 | 3.0 | 15.0 |
| Environmental-structuring | 3 | 9.27 | 12.0 | 0.26 | 3.0 | 15.0 |
| Task strategy | 4 | 12.49 | 16.0 | 0.33 | 4.0 | 20.0 |
| Time management | 3 | 9.75 | 12.0 | 0.26 | 3.0 | 15.0 |
| Help seeking | 3 | 8.94 | 12.0 | 0.23 | 3.0 | 15.0 |
| Persistence | 3 | 10.49 | 12.0 | 0.28 | 3.0 | 15.0 |
| Self-evaluation | 3 | 9.66 | 12.0 | 0.26 | 3.0 | 15.0 |
| Perception | 4 | 11.58 | 6.0 | 0.40 | 4.0 | 20.0 |

Note. Survey items were constructed with a 5-point Likert scale ranged from 1 strongly disagree to 5 strongly agree.

**Table 6.** Descriptive Analysis of *t* Test Table.

| | *n* | Mean | Std Deviation | Variance |
|---|---|---|---|---|
| **Pretest** | 120 | 88.07 | 8.07 | 65.1249 |
| **Posttest** | 120 | 83.22 | 7.22 | 52.1284 |

**Table 7.** Paired Sample *t* Test Table.

| Pair1 Pre & Post | Diff. Mean | Std. Deviation | Std. Error Mean | Conf. Lower | Conf. Upper | t | df | Sig. (2-Tailed) |
|---|---|---|---|---|---|---|---|---|
| | 4.86 | 2.83 | 1.141 | 2.54 | 7.18 | 4.178 | 119 | 0.000 |

## 5.4. Survey Qualitative Analysis

The qualitative analysis of the students' responses to the eleven-item, open-ended survey was performed through thematic analysis of the texts. All the data collected from the survey was in the form of written feedback. Thematic analysis of the text was employed to explain and describe the students' perceptions of the adaptive learning system (ALEKS) after using it in a semester. This survey intended to answer the second research question: What are the students' perceptions of using ALEKS as their adaptive learning system? The following section presents the results of the findings of each open-ended question in the survey.

Item # 1. Did you contact the instructor or help center if you faced a problem? If yes, how often? If no, why?

This item is intended to learn about the students' *help seeking* skill. Out of the total answers, 82% of the students mentioned they never or rarely sought help while working with ALEKS. They noticeably stated that they did not adopt help seeking approaches as a habit to improve their SRL learning skill experience. The students also mentioned that they mainly sought help just to check out the due dates or assignments.

Item # 2: How persistently did you check the ALEKS options (such as notifications, grade book, or progress charts)?

This item is intended to learn about the students' *persistence* skill to check the system notifications frequently. Only around 36% of the students mentioned they persistently checked the system's options (such as notifications, grade book, or progress charts). The thematic analysis of this question showed that almost two-thirds of the students were not persistent enough to check the system options regularly to find out their learning status or they even were not aware of the availability of such options.

Item # 3: Did you take notes or write annotations when doing tasks or reading materials in ALEKS?

A study by [32] showed that highly self-regulated learners made better use of resources and tools by taking notes of relevant information than students with lower SRL skills. Therefore, notetaking was used as a *task strategy SRL* skill in this study to find out if the students preferred to use this skill. Out of the total number of participants, 66% indicated that they took notes while studying in ALEKS which shows a high potential of ALEKS to scafold this skill in students.

Item # 4: How did you manage and structure the learning environment of ALEKS based on your preference and how often? For instance, using the calculator, periodic table, data, and formula, change the setting, use the test help button (elaborations), use the progress charts, etc.

How an online learning environment is structured or organized can bring a positive attitude toward the system [33,34] and, consequently, can lead to higher self-regulated learning. For this study, item#4 asked about students' willingness to *structure their ALEKS learning environment*. ALEKS provides some optional tools that students can add to their course dashboard to structure their learning environment based on their preference. For example, they can add a calculator or the periodic table to their dashboard if they feel this is necessary and can facilitate their learning. Or they can use the help button to receive more scaffolding and elaboration about the test items. Out of the total answers, 81.2% of the students mentioned they changed the ALEKS setting based on their preference at least one time while working with ALEKS. Most of this group indicated they used the help button in the tests more than any other options to receive more explanation for each test item.

Item # 5: How did the design of ALEKS line with your goal setting?

*Goal setting* is a prominent factor affecting self-regulated activities [35]. Furthermore, goal setting involves learning through affective reactions such as higher self-satisfaction when goals are reached [36] and can motivate learners to put forth a higher effort and persistence [37]. Almost half of the students (47.05%) mentioned that the system helped them set and obtain their learning goals. The general theme of the students' responses to this question was that multiple practices and repetitions in ALEKS were the main reason to attain their learning goals.

Item #6: How did you know if you were successful in learning Chemistry with ALEKS? How could ALEKS help you to understand how much you were successful?

This question was intended to obtain information on students' *self-evaluation* skill.

ALEKS provides different forms of reports and evaluation tools for the students to check their achievement, eliminating the need for any further evaluations by instructors. These reports include grade book, pie graphs, progress bars, knowledge checks, to name a

few. To answer this item, over 90% of the students mentioned they used at least one of the evaluation tools in the system to evaluate their success in the course.

However, the students' perspectives toward their success or failure in this course ranged from positive (58.2%) to negative (41.8%) views. While some of the students believed the system helped them to be successful in the Chemistry course, others found themselves unsuccessful.

Item # 7: Generally, what was your perception toward the system? Did the system support your learning? Why?

The next question in the survey, geared toward investigating the students' *Perceptions* toward the system. Out of the total number of responses, 53% indicated they were not satisfied from the system, and it hardly supported their learning. This group of students believed that ALEKS was complicated, frustrating, and time consuming. Additionally, the lack of proper instructions in the system, complex explanations of topics, problems, tests, and hard penalties (docking points for minor mistakes) were among the reasons they felt the system did not support their learning. On the other hand, 47% of the students pointed out that ALEKS supported their learning for its fast pace of learning, practicing test problems, and repeating complicated concepts.

Item #8: How could ALEKS help you to manage your time?

Out of total number of responses, 66% believed the system was time consuming but around 34% noted the system helped them to increase *time management* skills. It seems that this adaptive system needs students with high time management skills to effectively prioritize the assigned tasks and assessment and meet the predetermined objectives on time without procrastination.

Item # 9: What are the benefits of the ALEKS system?

By analyzing the students' answers to this question, varied perceptions about the system's *benefits* were identified. The students mentioned that ALEKS could help them manage their learning better. Or it could help them practice the test items more, receive quick feedback on their performance, and manage their time more efficiently. Another benefit of ALEKS mentioned by the students was that the learning experience was enjoyable for many reasons, such as improving comprehension of the subject, a sound tracking system, different ways to solve a problem, and an individualized learning pathway.

Item # 10: What are the hurdles or limitations of the ALEKS system?

The students believed that ALEKS had some *hurdles and limitations*. The main theme found in their responses about the system's hurdles was the complications while working with it. They pointed out that the system had poor instructions, confusing explanations, hard penalties, tough objectives to reach, no detailed explanations of students' errors, and a confusing structure and layout.

Item # 11: What advice would you give to other students planning to take similar courses?

The last question of this survey was to gather informative *suggestions* from current students for future ones using ALEKS. New students should be aware that ALEKS probably takes more time than any other online systems, such as Moodle or Bblearn, to understand and to complete the assigned work. Therefore, they should be prepared for this fact and be cautious of their workload. Students need to become sufficiently familiar with ALEKS at the beginning of the course to better understand how to solve problems, or which feature helps them most with their learning. Instructors can share these suggestions with other students working specifically with ALEKS to inform them in advance of the current issues of the system and how to deal with them better and also how they can be successful. In this way, new students can become more prepared to start their learning journey and tackle the system complexities. Table 8 shows the key themes of the survey with some of the students' quotes elaborating each.

**Table 8.** Key themes of the survey with students' quotes (*n* = 120).

| Theme | Student Quotes |
|---|---|
| **Help seeking** | <ul><li>I did not face problems that I could not solve by myself.</li><li>I did not care that much.</li><li>I did not even know these options even exist.</li></ul> |
| **Persistence** | <ul><li>I just checked the system when there was a notification alert.</li><li>I did not check them frequently.</li><li>I did not even know about [their] existence.</li></ul> |
| **Task strategy** | <ul><li>I took notes when the content was challenging.</li><li>I used notes as a future reference.</li><li>I took notes to get a better understanding of what I was doing.</li></ul> |
| **Environmental-structuring** | <ul><li>The calculator and periodic table are very accessible, and I used them in ALEKS.</li><li>I used the help button a lot in the tests.</li><li>I changed the setting of my ALEKS dashboard by adding my progress charts on my first page to see how well I performed in my test.</li></ul> |
| **Goal setting** | <ul><li>The repetition of materials helped to gain my goal.</li><li>I matched my weekly schedules based on ALEKS weekly goals.</li><li>ALEKS design made it easier to set my learning goals.</li></ul> |
| **Self-evaluation** | <ul><li>Looking at my grade book kept me in the loop of my success.</li><li>I check the reports page.</li><li>I checked the progress bar to see how much I could proceed.</li></ul>**Positive**:<ul><li>When the objective showed I could get through it, I felt I was successful.</li><li>I felt successful because I could finish the objectives of each topic.</li><li>Constant reminders of congratulations on learning topics from ALEKS.</li></ul>**Negative**:<ul><li>The ALEKS system is not an accurate system to show how well I know the content.</li><li>I did not [know if I was successful]. Even if you make one simple mistake, you have to redo the problem, even though you know the content.</li><li>I did not [know if I was successful]. The system did not consider minor errors in calculations and deducted an entire point for a small mistake.</li></ul> |
| **Perceptions** | <ul><li>It did not. It made things more confusing and challenging.</li><li>It was time consuming.</li><li>It made me to focus on completing assignments quickly than fully digesting the material.</li><li>It allowed me to expand my knowledge in problems similar to those in the tests.</li><li>It allowed alternative ways of solving problems.</li><li>It provided more support and questions for concepts that I struggled with.</li><li>I learned how to use the equations properly and analyze how things will change due to different factors.</li></ul> |
| **Time management** | <ul><li>I could manage my time correctly based on the weekly objectives of the course.</li><li>It needs more time for understanding key points.</li><li>It took up too much time, and I felt like I could never finish the objectives in ALEKS.</li></ul> |
| **System Benefits** | <ul><li>Repetition of wrong answers helped to understand better.</li><li>It allowed me to work on each problem several times.</li><li>I got instant feedback on my homework.</li><li>I did enjoy the tasks and assessments as they forced me to practice the issues through different ways of asking the question.</li><li>I think it did improve my comprehension of Chemistry a lot.</li><li>I like its tracking progress.</li><li>I enjoyed individual learning with immediate feedback.</li></ul> |

**Table 8.** *Cont.*

| Theme | Student Quotes |
|-------|----------------|
| **System hurdles** | • Instructions were very misleading.<br>• Extra steps, repetitive unimportant detailed points, annoying tasks, and so specific for absolutely no reason.<br>• ALEKS layout was confusing.<br>• Problems were too complicated for an intro-level chemistry course.<br>• Unfriendly, defeating, and homework-based system. |
| **Students' suggestions** | • Time management ahead of time.<br>• Ask for help when you need it.<br>• Use all the available materials.<br>• Stay on top of objectives.<br>• Pay attention to instructions.<br>• Do more practice.<br>• Expect frustration.<br>• Use ALEKS helpful comments and feedback. |

## 6. Discussion

Numerous studies showed the important role of SRL skills in students' success in online settings [38,39]. Web-based online learning requires developing SRL skills in learners which is not possible without understanding and paying attention to their characteristics and to develop strategic learning skills. The SRL skills are important in the online learning environments which require autonomous and independent learners [6,40,41]. Adaptive learning, as a form of online and distance education, is not an exception. In this platform, there is a vital need to develop independent and autonomous learners, because students are alone with the system, and they do not have any interactions with others. Therefore, they not only need the SRL skills to work independently in advance but also, they should be provided with the opportunity to enhance and strengthen these skills while working alone with the system.

There is a value in studying and analyzing these crucial skills in adaptive learning platforms and finding ways to help students improve their SRL skills more than even before with the increasing application of this learning system at universities.

The following section presents discussions of the main results gathered from the questionnaire and survey, including influential factors in decreasing SRL score, the need for SRL skills improvement, and ALEKS benefits and hurdles.

### 6.1. Influential Factors in the Drop of SRL Skills' Score from Pretest to Posttest

The result of the quantitative analysis showed a significant drop in SRL skills' score. This drop can be due to multiple reasons such as students' motivation, boredom, and dissatisfaction. These factors are discussed in the following.

It seems that ALEKS, like many innovations, could bring high motivation to students using the system for the first time because it is a new, engaging, and individualized learning courseware. Such novelty could have resulted in the higher self-reported scores of students' SRL skills in the ASRQ at the beginning of the semester. As [41] noted, motivation greatly influences self-regulated learning skills. However, after four months of working with ALEKS, the students better understood its complexities, leading to a decrease of their motivation and, probably, a lower rating of their SRL in the posttest. Therefore, the students' initial motivation for working with this new system tapered as the semester came to an end; this could be a reason behind the drop of the SRL skills' score.

Another factor that could have contributed to the drop in SRL score in the posttest can be the student's boredom and dissatisfaction with the system. ALEKS provides an isolated learning pathway. This isolation can increase boredom, impairing learning and performance [42], and decrease the satisfaction toward this learning environment. Students' satisfaction with their learning environment affects their SRL skills [43]. Therefore, it can be

assumed that the students' increasing boredom and dissatisfaction with the system could be another reason behind the decrease in the SRL skills' score.

### 6.2. The Need for High SRL Skills in Adaptive Students

Both qualitative and quantitative results of this study unanimously indicated the importance of increasing SRL skills in adaptive students. As mentioned earlier, online learning environments require higher levels of SRL for autonomous learning [44]. One of the requirements of being an independent and autonomous online learner is to have high SRL skills [20]. Therefore, due to similarities between online learning and adaptive learning, the need for independent learners with a higher level of self-regulation is indispensable in the isolating environment of adaptive learning. Thus, students planning to work within the ALEKS setting need to be scaffolded to develop SRL skills for independence and self-sufficiency through embedding articles, videos, ASRQ questionnaire, surveys, activities, or games demonstrating and fostering SRL skills in the system.

As shown in Figure 1, self-evaluation (90%), environmental-structuring (81%), and task strategy (66%) were the highly reported skills in the survey by the students. The other skills reported less than (50%). The following section specifically discusses the results obtained from the survey regarding each of these SRL skills.

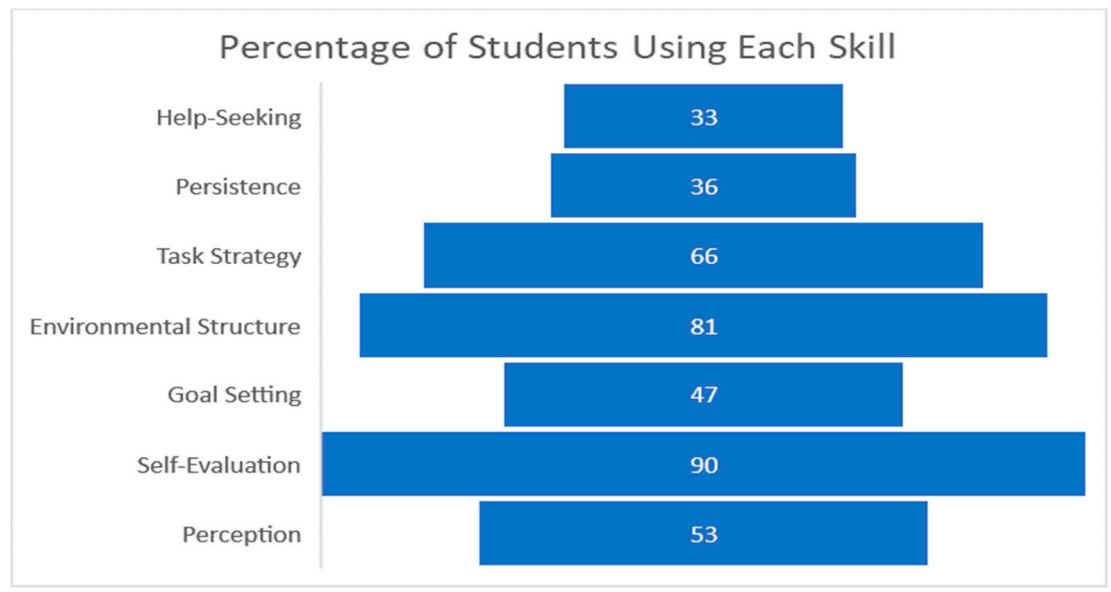

**Figure 1.** Percentage of students who reported using each skill included in the study.

#### 6.2.1. Help Seeking

It seemed that the lack of help seeking was mainly because, in this specific class, students had face-to-face sessions and in-person meetings with the instructor, which enabled them to solve their problems with the help of the instructor and their peers. However, in fully online adaptive courses (without in-person sessions), students usually do not have access to anyone except the system Help Center. As it stands from students' responses, ALEKS Help Center was not particularly strong to provide the appropriate knowledge-based help to students. Hence, it is suggested that ALEKS is equipped with efficient help options such as a 24/7 chatbot, live chat box, or online learning groups who fully are aware of the course, syllabus, and content to encourage students to seek help when needed.

#### 6.2.2. Persistence

One of the points that could be seen in many responses was that students were not even aware of the system's available options. This lack of awareness of tools in the

system should be alarming for instructors working with ALEKS. To provide more student awareness, instructors could introduce the adaptive courseware in detail at the beginning of the course and not just rely on a short system tour available in ALEKS. Instructional designers can embed some mandatory videos in online-only adaptive courses to introduce the system's help options. This, on the other hand, can increase students' persistence to check the system's helpful options (like progress bar or grade book) and become aware of their amount of success in the early stages of the course.

### 6.2.3. Task Strategy

A high percentage of using notetaking as a task strategy showed that students using the adaptive system in this study had a high potential to practice notetaking skill for different purposes of the course (like tests or assignments). Therefore, it is suggested that ALEKS embed a notetaking tool within the system with the ability to be saved, retrieved, and edited to enable students to write down their notes or annotations and use them later.

### 6.2.4. Environmental-Structuring

There are only a few options available to add/use in the system for structuring the learning environment, but these options are very useful and practical. In this regard, the willingness of students to use ALEKS options suggested that ALEKS embed connected tools to enable students to structure their online learning environment based on what they think best suits their needs. Some adaptive features available in other online systems include adaptive navigation [45], which supports learners by changing the appearance and structure of navigations; or adaptive social context [46], which provides customized social interactions/activities based on students' interests.

### 6.2.5. Goal Setting

ALEKS is mainly designed for math-based courses (like Chemistry, Physics, or Mathematics). Therefore, the numerous activities for practices and repetitions in the system are crucial to help users reach their learning goals, i.e., being master in solving the problems, memorizing, and applying the formulas, and being successful in tests. ALEKS can be helpful to increase students' goal setting skill by providing opportunities for each learner to set their own learning goals and achieve those goals at their own pace. ALEKS can also provide supplementary materials to help students achieve their learning goals based on their success or failure during the previous learning steps like informative videos, tutorials, or topic-based goal-setting activities.

### 6.2.6. Self-Evaluation

The responses of students to their amount of success in this course showed a range of negative and positive feedback. More than a half of them found themselves successful; however, there were some complaints about their feeling of achievement. One of the main complaints was a strict penalty for simple mathematical mistakes or missing a test item. Students should answer three test items correctly in a row to successfully proceed to the following knowledge level. For example, if a student does not know a single content in a test item, they lose the point of that specific item and one of their earned points from the previous correct answers. Therefore, any minor mistake, even in calculation, pushes them one step back. This process seems to be complicated, tedious, and creates frustration and disappointment for students, especially for weaker ones. This is because students with inadequate knowledge levels need to study more, do more assignments, take more tests, and spend more time in the system to reach the same level as the stronger students—and also achieve the course predetermined objectives. Another point against students who need more remediation is that ALEKS is mainly used in academic institutions with set semester schedules, i.e., all students, regardless of their knowledge level, should finish the course in one semester. Consequently, it is suggested that when higher education institutions use

ALEKS, they consider this issue and accommodate the needs of students in need of more remediation, so they have more time to finish the course than a single semester.

### 6.2.7. Perception

One factor influencing learners' self-regulated learning is the course quality in online learning. When students face some issues managing their online learning, this may decrease their SRL and finally hinder their success [47]. The most frequent comment toward ALEKS was about the system's complexity, which can reduce the quality of the course and, as a result, drop in SRL score. Therefore, this issue should be considered quickly by ALEKS designers.

The students in ALEKS have the autonomy to finish each module as quickly as they can (It should be mentioned that in the current institutional calendar, students still should take the final tests in ALEKS at the end of the academic semester, and they cannot finish the course prior to the final week determined by the corresponding institute). Stronger students mostly enjoy this feature because they just study unknown topics without needing to repeat the known materials, which is a completely different experience from academically weaker students. ALEKS, furthermore, provides numerous exercises, for each topic, similar to those in tests; therefore, students become ready for the tests. As mentioned before, the instructional design of ALEKS is primarily helpful in courses such as Mathematics or Physics, full of formulas and problems. The repetition of formulas in different test questions helps students memorize them and use them appropriately. However, this might increase the rote memorization to pass the tests and not learn the topic deeply. Therefore, instructional designers can add more higher-order thinking tasks and activities (such as design an experiment, concept mapping, or cooperative learning) to help students learn more deeply.

### 6.2.8. Time Management

Adaptive learning aims to adjust the learning assignments to a busy home and work schedule, especially during the lack of direct instruction such as during the COVID lockdown. Adaptive learning customizes the learning materials based on each student's knowledge level and provides relevant content and appropriate remediation. This can help students to manage their time correctly. However, it seemed that the lack of instruction on managing the time while working in the system and the complexity of the topic added up to the system's new features overwhelmed the students of this study and they spent a lot of time dealing with the system. While proper instruction at the beginning of the semester on how to manage the time can save a great deal of students' time.

### 6.3. Adaptive Learning Systems Benefits and Hurdles

The limitations and benefits mentioned in this study are crucial in students' success and should be considered by instructional designers. Since ALEKS provides a personalized but isolated learning environment for students, the instructions, elaborations, guidelines, procedures, and objectives can be designed in a more user-friendly mode and more straightforward language to be understandable for all users. The participants in this study unanimously believed that ALEKS is a complex system. Therefore, future students using adaptive learning systems need to take courses that do not have such demanding conceptual learning loads, and which have fewer assignments and tests. To alleviate some of the complexities of the system, instructional designers can:

- Embed audiovisual tools for learners with different learning styles.
- Integrate higher-order thinking tasks instead of multiple-choice tests.
- Reduce the number of topics in each course.
- Give a chance to students to make mistakes without losing any points.
- Remove the repetitive procedures, tests, or tasks.
- Add more help options.
- Scaffold students to improve their SRL skills.

- Provide virtual tours of different features of the system and benefits of each.
- Embed collaborative group activities and interactive social tools (like a chatbot, social media, etc.).
- Create a more flexible setting with a more straightforward language.
- Provide some audiovisual adaptive learning materials (instead of adaptive tests) based on the knowledge level of students.

Adaptive learning systems in general, and ALEKS in particular, provide a unique learning environment for each learner, which is difficult to find in other online courseware. Usually, adaptive learning systems are ideal courseware to teach formula-based courses such as Mathematics, Statistics, or Physics. The repetition of topics, consecutive assessments, and diagnosis of each students' knowledge level are also some of the unique features of these systems. They also facilitate super-effective learning by offering the students a selection of the topics they are currently ready to learn. ALEKS, in particular, provides a trove of real time, detailed reports providing educators and instructor with clear, up-to-date information to observe how their students' progress [48].

## 7. Implications

The ASRQ and survey instruments in this study are valuable tools that adaptive courses' instructors can use. If instructors administer the ASRQ at the beginning of the semester, they can understand students' level of SRL skills at the early stages of the semester. This will enable them to modify the course and accommodate the possible students' needs in an ALEKS-based course. Additionally, students' SRL skills in each of the eight factors of the ASRQ can shed light for instructors to understand how well students can manage their time, set goals, structure their learning environment, evaluate their learning, use task strategies, seek help, and be persistent in the individualized learning environment of adaptive systems. The survey also can be used by instructors to assess students' perceptions at different points of the course and avoid continuing frustration or confusion for the entire semester. As indicated in previous research, there is a positive relationship between learners' perception of online courses and their academic achievement [6]. Thus, perception is a determining factor in students' achievement, which can be considered by administering this survey when teaching adaptive courses. Instructional designers can embed these instruments in adaptive learning systems to understand the benefits and challenges of an adaptive environment through the eyes of its users and find better solutions for the existing issues and improve the system's usefulness in a timely manner.

Instructors can also refer to the results of this study and try to reinforce the skills that students are reported weak compared to the other skills, for instance, help seeking, persistence, and goal-settings. Instructors of online adaptive courses can embed some extra activities into the course shell to teach students how to set goals for their daily class activities to be successful. Or, they can embed videos explaining different available options in the system, keeping them on track and increasing their persistence. Additionally, they can inform the students how to get connected to the help center of the system and reach out to solve their technical problems on a 24/7 basis. On the other hand, the result of this study indicated that students are stronger in self-evaluation, environment structuring, and task strategy skills. This information can be used as a turning point to encourage students to improve these skills to be successful in their adaptive learning environment by introducing these skills and improving them via providing articles, videos, games, or activities.

The results of this study can help both instructors and instructional designers who are working hard to eliminate the glitches of adaptive systems and to improve them constantly.

## 8. Limitations

This study was delimited to the students at a university in the Southwestern part of the U.S. who enrolled in the Intro to Chemistry course equipped with the ALEKS adaptive learning system. Therefore, the findings of this study may not apply to students in other fields of study or different systems. Moreover, the ASRQ and survey collected self-reported

data that might contain biases, misrepresentation, lack of consideration in answering the questions, or proper understanding of the questions [49,50].

The other limitation of this study was the sample size. As mentioned before, the sample size of the students who participated in both pretest and posttest ($n = 120$) was not large enough due to the limited number of adaptive courses available in this higher education institute. Consequently, the results of the study may not be generalizable to a larger population that is not comparable to the population included in the current study.

Another limitation was that this research mostly relied on the total score of eight SRL variables in the questionnaire, so it did not study each SRL factor individually. It should also be mentioned here that this system was not specifically designed to support the development of SRL skills, and the researchers could not control the course design.

## 9. Suggestions for Further Research

Researchers in future studies could focus on a large, diverse sample of students in different adaptive courseware to allow for more analysis regarding the effectiveness of this new environment on students' SRL skills. Replica studies using the ASRQ followed by a structured follow-up interview with students and affiliated faculty in diverse adaptive learning courses should strengthen the results of this study. The degree of change for each SRL skills also needs further research. It is also suggested that researchers accomplish further studies to understand the predictive utility of each SRL variables of students' final grade. Finally, the influence of different subject matters and adaptive courseware on SRL skills can be investigated to shed light on the findings of this study.

## 10. Conclusions

Today's society needs citizens who are creative, social, and integrative thinkers. Adaptive learning systems need to develop learners with effective SLR skills (affect, emotion, self-regulation, higher-order thinking, etc.) to deal with complexities of the society, to be able to create social and collaborative work environments, and to know how to be lifelong learners in a non-patterned, complex, and ambiguous network of connections [38]. This research study extended existing literature via exploring the changes of SRL skills' score in an adaptive learning environment. Students' total SRL scores in pre- and post-test were compared, and the changes over time were calculated by running a dependent sample $t$ test. This study concluded that the total SRL score dropped significantly after four months of working with the ALEKS adaptive system. Additionally, the result of the qualitative part of the study indicated that some issues, such as lack of motivation and social presence of student; moreover, complexity, dissatisfaction, boredom, or hard penalty might be the reasons behind the drop of SRL skills at the end of the semester. Additionally, it was found that some SRL variables including self-evaluation, environmental-structuring, and task strategy were the highly used skills among the adaptive students in this study.

It is assumed that adaptive learning is likely to increase rapidly worldwide at different levels of education. Soon, this technology will become widespread, and its educational trends, principles, and standards will be widely used in all educational institutes [3]. Adaptive technology needs autonomous and independent learners with high SRL skills, and social trends demand it. As previous research shows [51], SRL skills need to be improved in all online settings, and adaptive learning is also a newer format of online learning. Therefore, equipping the adaptive systems with the best options, materials, and procedures to motivate students to retain and improve SRL skills is highly needed. Moreover, instructors and instructional designers must work collaboratively to find and solve the glitches of the next generation of online learning, i.e., of adaptive learning.

**Author Contributions:** Conceptualization, H.H.; Data curation, H.H., S.J.W.A. and C.-J.Y.; Formal analysis, H.H.; Investigation, H.H.; Methodology, H.H.; C.-H.T. and C.-J.Y.; Validation, H.H., C.-J.Y. and S.J.W.A.; Writing—original draft, H.H. and L.S.-M.; Writing—review & editing, All authors have read, edited, and agreed to the published version of the manuscript.

**Funding:** This research received no external funding.

**Institutional Review Board Statement:** The study was conducted according to the guidelines of the Declaration of Helsinki, and approved by the Institutional Review Board of Northern Arizona University, Protocol Code: 10999622-2, Date of Approval: 19 December 2017).

**Informed Consent Statement:** Informed consent was obtained from all subjects involved in the study.

**Data Availability Statement:** The data are not publicly available due to privacy issues.

**Conflicts of Interest:** The authors declare no conflict of interest.

### Appendix A. ASRQ Questionnaire

#### Goal setting

1. I set academic goals for my adaptive courses.
2. I create a study plan for my adaptive courses.
3. I track my progress in my adaptive courses.

#### Environmental-structuring

4. I choose a certain amount of time to study for my adaptive courses.
5. I choose a special place to study for my adaptive courses.
6. I avoid any distractions when I am studying for my adaptive courses.

#### Time management

7. I have a specific schedule to study for my adaptive courses.
8. I allocate specific studying time for my adaptive courses.
9. I use my time efficiently to finish my exercises in my adaptive courses.

#### Help seeking

10. I contact the 'Help Center' to solve my technical problems in my adaptive courses.
11. I use 'Tutorials' and/or 'Help Page' to solve my technical problems in my adaptive courses.
12. I contact the instructor and/or knowledgeable peers to help me solve problems with content in my adaptive courses.

#### Persistence

13. I make an extra effort to complete difficult exercises in my adaptive courses.
14. I am Persistent in working on topics that I have not learned in my adaptive courses (Note: ALEKS indicates your mastery level in each topic).
15. I do not give up until I finish all the exercises in my adaptive courses.

#### Self-evaluation

16. I evaluate the usefulness of the learning strategies that I use in my adaptive courses.
17. I evaluate my performance in my adaptive courses every time I login into the system.
18. I study the materials more than once to figure out my problems in my adaptive courses.

#### Task strategy

19. I use a variety of learning strategies in my adaptive courses.
20. I manage the content and technology challenges in my adaptive courses.
21. I fill-in my knowledge gaps in the subject matter by using the adaptive learning system (Note: the ALEKS system).
22. I try to take more notes because they are more important for learning in the adaptive course than in a regular classroom.

#### Perception

23. I feel my adaptive courses are engaging.
24. I am confident in the level of my knowledge in my adaptive courses.

25.　I have a positive learning experience in my adaptive courses.

26.　The system feedback meets my expectations.

### Appendix B. Survey

Item # 1: Did you contact the instructor or help center if you faced a problem? If yes, how often? And if no, why?

Item # 2: How persistently did you check the ALEKS options (notifications, grade book, or progress charts)?

Item # 3: Did you take notes or write annotations when doing tasks or reading materials in ALEKS?

Item # 4: How did you manage and structure the learning environment of ALEKS based on your own preference? For instance, using the calculator, periodic table, data and formula, change the settings, read the tutorials, use the help button, use the progress charts, etc.

Item # 5: How did the design of ALEKS line with your goal setting?

Item #6: How did you know if you were successful in learning Chemistry with ALEKS? How could ALEKS help you to understand how much you were successful?

Item # 7: Generally, what was your perception toward the system? Did the system support your learning? Why?

Item # 8: How could ALEKS help you to manage your time?

Item # 9: What are the benefits of the ALEKS system?

Item # 10: What are the hurdles of the ALEKS system?

Item # 11: What advice would you give to other students planning to take similar courses?

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
