# Peer review of "Assessment and Learning in Knowledge Spaces (ALEKS) Adaptive System Impact on Students’ Perception and Self-Regulated Learning Skills"

_education, doi:10.3390/educsci11100603_

Round 1

Reviewer 1 Report

Reviewer's summary of the study after reading the manuscript:

A learning course integrated with the Assessment and Learning in Knowledge Spaces (ALEKS) was used to collect data for this research, which shows the findings of applying the Adaptive Self-regulated Learning Questionnaire (ASRQ). The pretest and posttest are used to determine whether or not the students' Self-regulated Learning Skills (SRL) have changed between the beginning of the semester and the end of it. The findings revealed a statistically significant decrease in students' SRL abilities while using ALEKS. Students' perceptions and feedback in an open-ended survey were used to determine the reasons for the significant drop in SRL skills. These reasons included a lack of motivation, system complexity, a hard penalty for failing to comply, a fixed learning structure, a lack of social presence, and a lack of system practicality.

Dear authors, thank you for your manuscript. I enjoyed reading it. Presented are some suggestions to improve it:

(1) Please consider modifying the title of the manuscript so that it is easier for potential readers to find your study. Please include the words "Assessment and Learning in Knowledge Spaces (ALEKS)" if it is possible.

(2) Please include a section to help the readers understand why Assessment and Learning in Knowledge Spaces (ALEKS) was considered by the authors to be the most suitable type of adaptive learning platform for the purpose of your study to analyze the effects on the students, compared to another type of adaptive learning platforms?

(3) Please include a section to discuss what were the challenges faced when working with a vendor such as ALEKS which only provided information in its dashboards and is unable to divulge further information about the inner workings of its actual algorithms or how it generates the personalized content knowledge maps for individual students? How did your team overcome those challenges? This would be very beneficial to the readers as they would be able to learn from your expert knowledge.

(4) To improve the impact and readership of your manuscript, the authors need to clearly articulate in the Abstract and in the Introduction sections about the uniqueness or novelty of this study, and why or how it is different from other similar studies.

(5) Please substantially expand the discussion section, and compare your results to the ones found in similar studies. In particular, please cite more of the journal papers published by MDPI.

(6) Some of the references cited are not yet properly formatted, e.g., in your citation number [4], there was only the name of the author and nothing else. For the references, instead of formatting "by-hand", please kindly consider using the free Zotero software (https://www.zotero.org/), and select "Multidisciplinary Digital Publishing Institute" as the citation format, since there are currently 46 citations in your manuscript, and there may probably be more once you have revised the manuscript.

Thank you.

Author Response

Reviewer 1

Thank you so much for your fruitful comments. I tried to revise the manuscript based on your comments. Here are the references to the comments:

  • Please consider modifying the title of the manuscript so that it is easier for potential readers to find your study. Please include the words "Assessment and Learning in Knowledge Spaces (ALEKS)" if it is possible.

Comment 1: I would like to add ALEKS full name to the title but the title will have more than 15 words which is not acceptable by the journal. I added (Assessment and Learning in Knowledge Spaces) to the Abstract. But I don't mind adding it to the title if the long title is OK with the reviewers and editors of the journal. This is my suggestion to change the title and include ALEKS there: 

Assessment and Learning in Knowledge Spaces (ALEKS) Adaptive System Impact on Students’ Perception and Self-Regulated Learning Skills

  • Please include a section to help the readers understand why Assessment and Learning in Knowledge Spaces (ALEKS) was considered by the authors to be the most suitable type of adaptive learning platform for the purpose of your study to analyze the effects on the students, compared to another type of adaptive learning platforms?

Comment 2: On page 7, the reason for selecting ALEKS as the adaptive system in this study is added

  • Please include a section to discuss what were the challenges faced when working with a vendor such as ALEKS which only provided information in its dashboards and is unable to divulge further information about the inner workings of its actual algorithms or how it generates the personalized content knowledge maps for individual students? How did your team overcome those challenges? This would be very beneficial to the readers as they would be able to learn from your expert knowledge.

Comment 3: This is really a good question and actually needs a paper to focus on. But as I was not responsible to work closely with the vendors. I had access to the courseshell and students' info in the system) and just the E-learning department of my university was responsible to collaborate with the vendor, my knowledge about this question is very limited. I can ask one of the E-Learning staff to help me and write a section to respond to this question, but it definitely takes time and needs more space to be added to this article. But I am open to any suggestions to accommodate this gap in the article. 

  • To improve the impact and readership of your manuscript, the authors need to clearly articulate in the Abstract and in the Introduction sections about the uniqueness or novelty of this study, and why or how it is different from other similar studies.

Comment 4: I added a few sentences to the abstract explaining adaptive learning and why it is unique. I also mentioned the reason of the limited number of studies available due to the limited number of institutes using this system for its high expenses. 

(5) Please substantially expand the discussion section, and compare your results to the ones found in similar studies. In particular, please cite more of the journal papers published by MDPI.

I added some MDPI citations to the manuscript. But I could not find any MDPI article related to the topic of this study.

(6) Some of the references cited are not yet properly formatted, e.g., in your citation number [4], there was only the name of the author and nothing else. For the references, instead of formatting "by-hand", please kindly consider using the free Zotero software (https://www.zotero.org/), and select "Multidisciplinary Digital Publishing Institute" as the citation format, since there are currently 46 citations in your manuscript, and there may probably be more once you have revised the manuscript.

I tried to use Zetero but I could not figure out how to work with it in such a short time. I tried to fix the references manually. Sorry about that.

The manuscript is edited by an English editor.

Reviewer 2 Report

Authors provide an analysis of the impact of an adaptive learning environment in student’s perception and SRL skills. The paper is well written, and the methods and results are explained in detail. The paper is relevant, and it contributes to the literature in adaptive systems and SRL. I just provide some comments to improve the article:

  1. The paper is very good framed, although I would expect more references that analyse how adaptive learning systems can enhance (or not) SRL if there are studies in other systems different than ALEKS. This could serve to better justify the novelty of your paper.
  2. Appendices I and II appear as Appendix A and B (with letters) in the paper. You should be consistent with this.
  3. Some details of the thematic analysis (4.3.2) are missing. You should comment the different themes used and the reliability of the coders (e.g., using kappa)
  4. You analyse SRL skills at global level in the t-tests. It would be beneficial if you report similar results skill by skill, as you highlight in 5.2 the decrease of some specific skills such as help seeking and environmental structuring
  5. I would suggest authors to further discuss how results of this study could be used for other instructors with other adaptative systems. The ASRQ and survey are themselves a contribution, but authors could also indicate what parts of the results may generalize or may be relevant to be considered in other scenarios.
  6. You could provide more references to compare your results with other works (in the last sections)

Author Response

1. The paper is very good framed, although I would expect more references that analyse how adaptive learning systems can enhance (or not) SRL if there are studies in other systems different than ALEKS. This could serve to better justify the novelty of your paper.

This is a very good suggestion. But this is a very novel study and the questionnaire used for this study has been developed and validated for the first time. And this paper is also the first result of applying this new questionnaire. There are other studies have been done regarding SRL and Online Learning that I cited in the paper and based on those research studies, I suggested SRL skills also are needed to be noticed and investigated in adaptive learning environments.

2. Appendices I and II appear as Appendix A and B (with letters) in the paper. You should be consistent with this.
Thank you so much for your catch.

3. Some details of the thematic analysis (4.3.2) are missing. You should comment the different themes used and the reliability of the coders (e.g., using kappa)
Thank you for noting this. I revised this part on page 9. I added K statistics and revised the part based on this comment

4. You analyse SRL skills at global level in the t-tests. It would be beneficial if you report similar results skill by skill, as you highlight in 5.2 the decrease of some specific skills such as help seeking and environmental structuring
This is a very good point and I suggested that in the further research. Right now, I’m working on a different paper focusing on the subskills of SRL. Due to different statistical measurements, purpose, research question, etc. I did not include that in this paper. But Im working on it. Thank you!

5. I would suggest authors to further discuss how results of this study could be used for other instructors with other adaptative systems. The ASRQ and survey are themselves a contribution, but authors could also indicate what parts of the results may generalize or may be relevant to be considered in other scenarios.
On page 22, implication section, I added a paragraph focusing on the application of the some of the results of the study. Also, in the implication, I already noted how ASRQ and survey can help AL instructors.

6. You could provide more references to compare your results with other works (in the last sections)
I added more studies to the manuscript. Also, I added some MDPI articles related and added those too

Thank you!